

# Space-based NO$_x$ emission estimates over remote regions improved in DECSO

Jieying Ding[1,2], Ronald Johannes van der A[1,3], Bas Mijling[1] and Pieternel Felicitas Levelt[1,2]

[1]Royal Netherlands Meteorological Institute (KNMI), De Bilt, the Netherlands
[2]Delft University of Technology, Delft, the Netherlands
[3]Nanjing University of Information Sciences and Technology, Nanjing, China

*Correspondence to*: Jieying Ding (jieying.ding@knmi.nl)

**Abstract.** We improve the emission estimate algorithm DECSO (Daily Emission estimates Constrained by Satellite Observations) to better detect NO$_x$ emissions over remote areas. The new version is referred to as DECSO v5. The error covariance of the sensitivity of NO$_2$ column observations to gridded NO$_x$ emissions has been better characterized. This reduces the background noise of emission estimates with a factor of 10. An emission update constraint has been added to avoid unrealistic day-to-day fluctuations of emissions. We estimate total NO$_x$ emissions, which include biogenic emissions that often drive the seasonal cycle of the NO$_x$ emissions. We demonstrate the improvements implemented in DECSO v5 for the domain of East Asia in the year 2012 and 2013. The emissions derived by DECSO v5 are in good agreement with other inventories like MIX. In addition, the improved algorithm is able to better capture the seasonality of NO$_x$ emissions and for the first time it reveals ship tracks near the Chinese coasts that are otherwise hidden by the outflow of NO$_2$ from the Chinese mainland. The precision of monthly emissions derived by DECSO v5 for each grid cell is about 20%.

## 1 Introduction

Nitrogen oxides (NO + NO$_2$ = NO$_x$) are important air pollutants and play a crucial role in climate change by catalyzing the formation of tropospheric ozone and forming secondary nitrate aerosol (IPCC, 2014; Shindell et al., 2009). Anthropogenic emissions are the main source of NO$_x$. Better emission estimates may help policy makers to mitigate the adverse influence of those air pollutants. However, it is difficult to obtain up-to-date emission data. Bottom-up emission inventories rely on gathering statistical data, which is time consuming and often resulting in a delay of a few years. In addition, statistics on emission-related activities and emission factors introduce large uncertainties in the bottom-up emission inventories. Satellite observations provide long-term NO$_2$ column concentrations with a global coverage and they are able to improve NO$_x$ emission estimates (Streets et al., 2013). Emissions of dominant sources like megacities can be derived from satellite observations using a simple mass-balance approach (Martin et al., 2003). Smaller sources, like biogenic emissions or ship tracks, especially if they are close to the dominant sources are more difficult to distinguish. These lower NO$_x$ emissions are derived from satellite observations with inversion techniques taking into account the transport of NO$_x$ (e.g (Miyazaki et al., 2012; Stavrakou et al., 2008)).

Mijling and van der A (2012) developed the inversion algorithm DECSO (Daily Emission estimates Constraint by Satellite Observation) based on an extended Kalman Filter with a regional chemical transport model (CTM). This algorithm takes the transport of NO$_x$ from its source into account. The emission estimates of DECSO improve upon existing emission inventories in the Middle East and South Africa (www.globemission.eu). The DECSO algorithm has the capability to show trends and seasonality on a provincial scale for China and the derived emissions are in good agreement with the bottom-up inventories in Asia (Mijling et al., 2013). Ding et al. (2015) improved the algorithm performance by updating the CTM and adding a satellite data quality control. The improved temporal and spatial resolution enables the detection of e.g. the NO$_x$ reductions during the Nanjing Youth Olympic games in August 2014.

Estimating emissions from satellite data is relatively new, and many developments are ongoing in this field. General weak issues in inventories are their low temporal resolution and the lack of information for small sources in remote areas. The



seasonal variability is normally poorly captured in emission inventories. Mijling and van der A (2012) showed that DECSO is able to detect the shipping emissions along the Chinese coastal seas, which are not included in the anthropogenic emission inventories INTEX-B (Zhang et al., 2009) and MIX (Li et al., 2015) (http://www.meicmodel.org). However, Ding et al. (2015) mentioned that emissions from DECSO have also large day-to-day fluctuations due to the noise on the observations.

In remote ocean areas where emissions should be close to zero, we still see relatively high emissions, which we refer to as background noise. This background noise is appearing in low-emission areas, where random negative emissions derived from the measurements are set to zero because the CTM cannot handle negative emissions. This results in, on average, a positively biased emission for this region. This background noise can have an unrealistic seasonal cycle. Because of this background noise, weaker emission sources, such as shipping emissions or small isolated power plants are difficult to

distinguish. A new version of DECSO, referred to as DECSO v5, has been developed with special attention for those issues. The improvement in observing low emissions will be demonstrated for two cases. One case is for biogenic sources and fire emissions in rural areas, which are relatively weak over often-large areas. The second case is for shipping emissions close to the coast, where low emissions exist in proximity to large sources of cities along the coast. Both emission sources have large uncertainties in existing inventories.

Estimates of biogenic emissions contain large uncertainties. In regions where fire emissions from forests and biomass burning are dominant, the seasonal cycle of $NO_x$ is not the same every year, since burnt areas are changing from year to year due to human activities (Schultz et al., 2008) and climate (Westerling et al., 2006). van der Werf et al. (2006) showed that the seasonality of fire emissions varied a lot from 1997 to 2004 in central Asia. He et al. (2011) studied biomass emissions in the Pearl River Delta region by using a bottom-up approach and showed that the uncertainties of $NO_x$ biomass burning

emissions vary from 80% to 220%.

Recent studies point out that ship emissions of $NO_x$ make a significant contribution to the total $NO_x$ budget. The global shipping emission inventories have large differences and high uncertainties from the statistical sampling of ship tracks, and the simplified parameterizations used in shipping emission calculations (Wang et al., 2008). Shipping emissions in e.g. the Indian Ocean, the Red Sea and the European seas are detected by SCIAMACHY, GOME, GOME-2 and OMI satellite

observations (Beirle et al., 2004; Franke et al., 2009; Richter et al., 2004; Vinken et al., 2014). Richter et al. (2004) mentioned that the signal of ship tracks towards China and Japan is lost in the transported $NO_2$ plume from the sources over land, so that it is difficult to estimate emissions by using the mass-balance approach. By using the improvements of DECSO described in this paper, we will show that DECSO is able to provide a $NO_x$ shipping emissions estimates near the coast of China and detect the seasonality of $NO_x$ emissions related to biogenic sources.

To improve the issues mentioned above, here we focus on performance of the DECSO algorithm over low-strength emission regions. We improve the error covariances of the sensitivity of $NO_2$ column observations to gridded $NO_x$ emissions. Adding an emission update constraint instead of using the satellite data quality control mentioned in (Ding et al., 2015) removes the unrealistic fluctuations of emissions. As a consequence, we use more information from the satellite observations. With the updates applied to DECSO, the algorithm is better able to detect shipping emissions and other small sources and shows a

more realistic seasonal cycle, especially over the low emission areas. In addition, we switch off the biogenic $NO_x$ emissions in the CTM to calculate total $NO_x$ emissions instead of only anthropogenic emissions. In section 2, we briefly describe the DECSO algorithm. The improvements are presented in section 3. We have applied the DECSO algorithm to the region of East China and for this area the improvements will be demonstrated. Monthly variability of emissions in different areas of East Asia and the shipping emissions over the China coastal region are shown in section 4, followed by a discussion of the

results in section 5.





## 2 Emission estimates with the DECSO algorithm

DECSO is an emission estimate algorithm for short-lived trace gases, which updates daily emissions on a mesoscopic scale with a spatial resolution of $0.25° \times 0.25°$. It constrains $NO_x$ emissions by combining the simulated $NO_2$ concentrations of a CTM with satellite observations. The essential part of DECSO is the calculation of the sensitivity of the $NO_2$ column

concentrations (on a footprint of the satellite) to the gridded $NO_x$ emissions, in which the transport of $NO_2$ over the model domain is taking into account. The inversion method used in DECSO is based on an extended Kalman filter.

We use the Eulerian off-line CTM CHIMERE v2013 (Menut et al., 2013) in DECSO. To run a simulation of CHIMERE, external forcings like meteorological fields, primary pollutant emissions and chemical boundary conditions are required. We have implemented CHIMERE on a $0.25° \times 0.25°$ horizontal resolution over the regions of Europe, Middle East, South Africa,

India and East Asia with 8 vertical layers up to 500 hPa. The model is driven by the operational meteorological forecast of the European Centre for Medium-Range Weather Forecasts (ECWMF).

The satellite observations in this study are measured by the Dutch-Finnish Ozone Monitoring Instrument (OMI) (Levelt et al., 2006) on NASA's EOS-Aura satellite. The spatial resolution is $24 \times 13$ km$^2$ at nadir and increases to about $150 \times 28$ km$^2$ at the edge of the swath. Its overpass time at the equator is around 13:30 local time. We use the tropospheric $NO_2$ column

data of the Dutch OMI $NO_2$ retrieval (DOMINO) algorithm version 2 (Boersma et al., 2011). The data set can be found on the Tropospheric Emissions Monitoring Internet Service (TEMIS) portal (http://www.temis.nl). We use the same data filter criteria for the DOMINO v2 data as mentioned in (Ding et al., 2015). The pixels affected by the so-called row anomaly (KNMI, 2012) and the four pixels at each side of the swath are filtered out. We use observations with a surface albedo lower than 20% and a cloud radiance fraction lower than 70%. The observations with clouds below 800 hPa are also excluded,

because cloudy and bright surface scenes have large influence on the quality of the retrieval product.

To compare the simulated concentrations from CHIMERE with satellite observed tropospheric columns, the modeled vertical profiles are extended from 500 hPa to the tropopause with a climatological partial column (2003-2008 average) simulated by the global CTM TM5 with a horizontal resolution of $2° \times 3°$ (Huijnen et al., 2010). We spatially average the simulated profiles on the model grid cell over the OMI footprints and apply the averaging kernel to the modeled profile. A

more detailed description of DECSO can be found in Mijling and van der A (2012) and Ding et al. (2015).

## 3 Improvement of DECSO

### 3.1 Observation and representativeness error calculation

The inversion in DECSO is based on the extended Kalman filter, which assimilates emissions using a CTM to relate the emissions to daily observed concentrations. The analysis of the emission $\mathbf{e}^a$ is calculated from the observed $NO_2$ column

concentrations $\mathbf{y}$ and the forecasted emission $\mathbf{e}^f$ at time t using equation:

$$\mathbf{e}^a(t) = \mathbf{e}^f(t) + \mathbf{K}(\mathbf{y} - H[\mathbf{e}^f(t)]),  \tag{1}$$

in which $\mathbf{K}$ is the Kalman gain maxtrix, and $H$ is the observation operator (translating emissions into column concentrations) which includes the CTM, the extension of the simulated column to the tropopause, the average kernel from the satellite retrieval, and the interpolation to the satellite footprint. The Kalman gain matrix $\mathbf{K}$ combines error covariances of

observations, model representation, and emissions to determine how much information from the observations is to be used in the emission analysis:

$$\mathbf{K} = \mathbf{P}^f(t)\mathbf{H}[\mathbf{H}\mathbf{P}^f(t)\mathbf{H^T} + \mathbf{R}]^{-1}  \tag{2}$$

$\mathbf{P}^f$ is the error covariance matrix of the forecasted emissions. $\mathbf{H}$ is the sensitivity matrix (Jacobian) describing how the $NO_2$ column concentrations on a satellite footprint depend on gridded $NO_x$ emissions, i.e. having matrix elements $H_{ij} = \frac{\partial y_i}{\partial e_j}$.

Typical dimensions of $\mathbf{H}$ are about 2500 (daily observations) $\times$ 16000 (emission grid cells).



**R** is the error covariance associated with observation operator *H*. The **R** matrix combines the observation error of the tropospheric $NO_2$ columns and the representation error introduced by the inaccuracy of the CTM and its projection onto the measured $NO_2$ column on the satellite footprint. **R** cannot be calculated directly, but we can derive its diagonal elements with help from the statistics of the Observation minus Forecast (OmF) of the $NO_2$ columns:

$$\sigma_{OmF}^2 = \sigma_R^2 + \sigma_{prem}^2 = \sigma_{obs}^2 + \sigma_{repr}^2 + \sigma_{prem}^2 \qquad (3)$$

The OmF variance consists of the variance of the **R** matrix ($\sigma_R^2$) (consisting of the observation error $\sigma_{obs}^2$ from the retrieval method and the representation error $\sigma_{repr}^2$) and the variance of the emission estimates propagated into the simulated column concentrations ($\sigma_{prem}^2$). $\sigma_{prem}^2$ is taken from the diagonal elements of matrix $\mathbf{HP^T(t)H^T}$.

In the previous versions of DECSO, the representation errors were calculated based on daily statistics of OmF since $\sigma_{obs}^2$ and

$\sigma_{prem}^2$ are known. The observation errors were initially taken from the observations and later described by an empirical exponentially relation with the observed column in version 3 (Ding et al., 2015). Mijling and van der A (2012) assumed that $\sigma_{repr}$ is linearly related to the simulated tropospheric column concentration, and derived this ratio from daily OmF statistics. However, from day to day the number of observations varies a lot due to variability in cloudiness and scenes of observations (e.g. ocean vs. populated land). In days with few observations, the statistics are insufficient to derive the proper

representation errors. This results in a lot of unrealistic day-to-day variations in $\sigma_{repr}$. In addition, because the observation coverage is smaller in wintertime than in summertime, the representation error is determined less accurate in winter due to the lack of statistics. Since emission updates (by equation (1)) are sensitive to the derived **R** matrix (used in equation (2)), this may lead to artificial seasonal fluctuations. We noticed that the average of **R** matrix elements is higher in winter than in summer leading to biased emissions in summer time.

To improve this issue, we study the OmF statistics for monthly and yearly periods and directly relate $\sigma_R^2$ to the $NO_2$ column instead of calculating the $\sigma_{obs}^2$ and $\sigma_{repr}^2$ separately. We run DECSO over East Asia (18−50 °N and 102−132 °E) for 2013. By binning the data in intervals of $1 \ 10^{15}$ molecules $cm^{-2}$ over the range of $NO_2$ column values, we use the OmF statistics and equation 3 to derive $\sigma_R$ per bin. For all observations 2013 we plot $\sigma_R$ against the average *c* of observed and modeled $NO_2$ column and find a linear relation (see Figure 1):

$$\sigma_R = \varepsilon_{rel}c + \varepsilon_{abs}. \qquad (4)$$

With relative coefficient $\varepsilon_{rel}$=0.55 and absolute offset $\varepsilon_{abs}$=0.20. Using data from other periods we found that this relation has very little variation from month to month and from year to year. Ding et al. (2015) used tuned synthetic error estimates for the satellite observations instead of the errors given by the retrievals. Tuning the error estimates of the satellite observations, as done for DECSO v3b, is no longer needed.

We apply this new linear relation in DECSO-exp. Figure 2 shows the comparison of yearly average emissions over the ocean for DECSO-exp and DECSO v3b. We see that the noise over remote areas is significantly reduced. The shipping emissions are smaller in DECSO-exp, but they are still clearly visible against the lower background values. The analysis of shipping emissions will be further discussed in section 4. The monthly emissions in 2013 over a selected remote ocean area (20 to 25 °N, 125 to 130 °E) derived using DECSO-exp are much lower than DECSO v3b (Figure 3). We see the emissions derived

by DECSO v3b vary from 4 to 10 Gg N per month with a summer peak. These emissions are too high for this area and the seasonal cycle is artificial. Since shipping emissions are the only emission source of $NO_x$ in this part of the ocean where few ships are passing, the emissions over this area should be close to zero. The emissions derived by DECSO-exp are 10 times lower in summer compared to the emissions of DECSO v3b. The artificial seasonality also disappeared, since the higher values of DECSO-exp in April, May and December are within the range of emission uncertainties. By using the linear

relation of $\sigma_R$ with the $NO_2$ column, the calculation of $\sigma_R$ is simplified, the background noise is reduced by 90%, from about 16,7 Mg month per grid cell to about 1.7 Mg month per grid cell, and the artificial seasonal cycle has been removed over low emission areas.





### 3.2 Emission update constraint

Occasionally large differences between observed and modeled $NO_2$ columns appear, which are probably related to wrong satellite retrievals during high aerosol events (Ding et al., 2015). Several studies pointed out that the DOMINO retrieval algorithm does not take the effects of high aerosol concentration properly into account (Leitão et al., 2010; Lin et al., 2014).

To reduce false emission updates due to day-to-day fluctuations, Ding et al. (2015) used an OmF criterion to filter out satellite data that show large differences compared to the model simulations caused by these high aerosol events. Although filtering out these satellite data solves the problem, it may also delete correct satellite data. For example, if a new power plant is built or an old one is removed, it can cause large changes of $NO_2$ concentrations in a single day. This is not simulated correctly by the CTM. With the OmF criteria, the emission updates will be unnecessary slow since a lot of satellite

data with large differences will not be used in the data assimilation. As we describe below, we improve the DECSO algorithm by using an emission update constraint to avoid unrealistically high fluctuations in the emission estimates.

In the DECSO algorithm, we use a persistent emission model, which assumes that the emissions of one day are equal to the emissions of the previous day, $e^f(t) = e^a(t-1)$. Assuming an optimal Kalman filter, the error of these emissions should have a Gaussian distribution, thus with 99.7% of emissions within the range of $3\sigma$ from the mean. Therefore, we will use a

$3\sigma$ interval to constrain the emission updates instead of the OmF criterion. To be able to handle both wrong observations due to e.g. high aerosol load as well as new emission sources, we apply the following procedure during the data assimilation process: in case that the emission update is larger than $3\sigma$, we regard it outside the emission range and set the emission update equal to $3\sigma$.

We run DECSO with this $3\sigma$ constraint for 2013 to test this new procedure, which is referred to as DECSO-exp2. With the

new constraint, we use 10% more satellite observations. Figure 4 shows that with the $3\sigma$ emission update constraint, the emission in Hong Kong is almost constant for the whole year with a little increase in winter. The emissions of DECSO v3b however are much higher (a factor of 2) in December than in January, which seems not realistic. Since the vehicle emissions are the main source in Hong Kong and the seasonality is obscure (Kwok et al., 2010). Over the ocean, the fluctuation of emissions becomes smaller with the new improvement as compared to DECSO v3b (Figure 3). Note that the R matrix

improvement is not applied in this run and therefore the improvement is only caused by the $3\sigma$ constraint, which can also be seen in Figure 3. With DECSO v4, the improvements on R matrix and the $3\sigma$ constraint are combined. The emission data over East Asia of this version are available on www.globemission.eu.

### 3.3 Logarithm of emission

Negative emission estimates in DECSO are set to zero, because the CTM is not able to handle negative emissions. As a

workaround, negative emissions are set to zero while the positive emissions in the nearby area are adjusted to compensate for the introduced bias. To make sure that the emission estimates remain positive, Brunner et al. (2012) replaced the emission state vector by its logarithmic value, assuming that the errors of this new state are Gaussian distributed. The assumption is not entirely correct in our case, but it is still a reasonable approximation.

To assess the effect of this approach in our setting, we define the logarithmic emissions ($\mathbf{x} = \ln(\mathbf{e})$) as the state vector in the l

Kalman filter. As a consequence, the observation operator $H$ (see equation 1) needs to be reformed. $H^l$ is the sensitivity calculator describing how the $NO_2$ column concentrations $\mathbf{C}$ depend on logarithmic emissions $\mathbf{x}$:

$$\mathbf{C} = H^l(\mathbf{x}) = H(\exp(\mathbf{x})) \tag{5}$$

In the Kalman filter, each element $H_{ij}^l$ of Jacobian matrix $\mathbf{H}^l$ of the new observation operator can be easily derived as shown by Brunner et al. (2012):

$$H_{ij}^l = H_{ij}\exp(x_j) \equiv H_{ij}e_j. \tag{6}$$





In addition to the Jacobian, we need the error covariance of the modeled logarithmic emissions. Based on the derivative of the logarithmic value of the emission with respect to the emission itself:

$$\frac{dx}{de} = \frac{1}{e} \tag{7}$$

the logarithmic emission error $q^l$ can be expressed as:

$$q^l = \frac{q}{e} \tag{8}$$

where $q$ is the error of the modeled emissions. Mijling and van der A (2012) gave an assumption for the dependence of the emission error $q$ on the emissions $e$:

$$q = \varepsilon_{abs} \exp\left(-\frac{\varepsilon_{rel}}{\varepsilon_{abs}} e\right) + \varepsilon_{rel} e. \tag{9}$$

$\varepsilon_{abs}$ (0.02 $10^{15}$ molecule $cm^{-2}$ $h^{-1}$) and $\varepsilon_{rel}$ (0.05) are the absolute and relative errors that are the dominating emission errors for respectively low and high emissions. By combining equation 8 and 9, we get the error for the logarithmic emissions. However, if the emissions are close to zero, the error will become infinite leading to computational errors. To avoid this, we define a new equation for $q^l$ to calculate the emission error, which leads to similar emission errors as in previous DECSO versions and avoids infinite error values:

$$q^l = \alpha_1 \exp\left(-\frac{\alpha_1}{\alpha_2} x\right) + \alpha_2 \tag{10}$$

By setting $\alpha_1 = 0.009$ and $\alpha_2 = 0.02$, the emission error $q = q^l e$ is in good agreement with the original $q$ calculated in equation 9. Using the new observation operator and the logarithmic emission error, we implement the logarithmic method based on DECSO v4 and run it for 2013, which is referred to as DECSO v4log. However, the results of this logarithmic method are not improved compared to DECSO v4. Instead, it increases the positive bias over low emission areas by a factor of 2. Also, the spin-up time is increased for low emission areas, which means it cannot detect the emissions changes on a small temporal scale. Since the improvements by using the new **R** matrix and 3σ constraint for emission updates already reduce the background noise and remove the artificial seasonal cycle, we decide not to use the logarithmic approach in this study.

### 3.4 The threshold of the sensitivity matrix and the effect of biogenic NO$_x$ emissions

A well-defined sensitivity matrix **H** (see equation 2) is essential for a good performing Kalman filter. As described by Mijling and van der A (2012), the calculation of **H** involves a simplified 2D trajectories to describe transport of the concentration plumes over the emission grid. In some cases the bulk of the plume is shielded by clouds and the satellite only monitors a fraction of the plume at its edge. The edge of the plume is usually characterized by low sensitivity matrix **H** elements resulting from trajectory calculations with a high uncertainty. The inversion becomes ill-conditioned, resulting in strongly fluctuating emission updates with high associated uncertainties. These emission updates are based on limited information from the satellite. Therefore we have set a minimum value for the **H** matrix elements. In the previous versions of DECSO, we used 0.05 hour as the minimum value. This value is still too low for the scarce winter observations at high latitudes leading to large biases in especially these locations. In this study we set the threshold value to 0.1 hour.

In DECSO v3b and v4, the derived NO$_x$ emissions represent anthropogenic emissions. Biogenic emissions are generated internally in CHIMERE v2013 following a parameterization (including land-use data) by the MEGAN model (Guenther et al., 2006). However, Li et al. (2007) showed that emission factors of soil NO emissions in the same type of forests can differ a lot. This introduces large uncertainties in estimated biogenic emissions. Figure 5 shows the ratio of biogenic NO emissions from MEGAN and anthropogenic NO$_x$ emissions from MIX. Biogenic emissions are about 5% of total anthropogenic emissions in rural provinces in summer. Unfortunately overestimation of the MEGAN biogenic emissions in those rural areas can introduce biases in the local anthropogenic NO$_x$ emission estimates by DECSO. For instance, in case MEGAN biogenic NO$_x$ emissions are overestimated, the NO$_x$ emissions derived by DECSO will be negative. As we mentioned in





section 3.1, DECSO sets negative emissions to zero and adjusts positive emissions in the surrounding area for compensation and mass balance.

By excluding biogenic $NO_x$ emissions in CHIMERE, we derive the total $NO_x$ emissions from satellite observations. Together with the improvement of **R** matrix, the 3σ constraint and the new threshold of the **H** matrix elements, this new

DECSO algorithm is referred to as DECSO v5 (the emission results calculated with this version is also provided on www.globemission.eu). The main differences of the various versions of DECSO mentioned in this paper are summarised in Table 1.

## 4 Results

We run DECSO v5 for the period 2012 to 2014 over East Asia. As we showed in section 3, the improvements in DECSO

reduce the background noise and removes artificial season cycles over low-emission areas. Due to this, it will enable better emission estimates over low emission regions like ship tracks in the ocean, small cities in deserts, grassland etc. Below we demonstrate the improvements in the emission estimates for these specific cases.

### 4.1 Seasonal cycle analysis

One way to validate the improvement of emission data derived by DECSO v5, is by comparing the forecast results of the

CHIMERE model used in DECSO v5 and DECSO v3b with OMI satellite observations over the model domain. We compared the average of $NO_2$ column per season per grid cell over the whole domain. The comparison shows that the improvement of DECSO v5 in wintertime (December to February) is significant. The modeled $NO_2$ columns of DECSO v5 are in much better agreement with OMI satellite observations in winter with a correlation coefficient (R) of 0.96 while R is only 0.87 for DECSO v3b. Yearly averages and the other seasonal averages of $NO_2$ columns of both versions are in good

agreement with OMI satellite observations, especially in summer. The R in summer of both versions is 0.97. The better agreement of DECSO v5 modeled columns with OMI satellite observations in wintertime results in a better seasonal cycle of emission estimates.

We compare the seasonal cycle of $NO_x$ emissions on a provincial level in 2012 with the MIX emission inventory of 2012 (the latest year available in MIX) over China and MIX emission inventory (Li et al., 2015) outside of China. We find that in

many provinces the seasonal cycles derived by DECSO are consistent with the MIX inventory. However, in most rural provinces, the difference between DECSO v5 and MIX is very large. Based on the ratio of biogenic emissions of MEGAN and anthropogenic emissions of MIX inventory shown in Figure 5, we divide provinces in China into two groups. We select a group of 13 provinces with large areas of relatively high biogenic emissions as rural provinces. The remaining 16 provinces in the domain are considered anthropogenic emission provinces. The R (correlation coefficient) of seasonal

emissions for anthropogenic emission provinces between DECSO v5 and MIX 2012 is 0.6, while for rural provinces the comparison is much worse (R is negative in most rural provinces) because MIX does not include biogenic $NO_x$ emissions.

Figure 6 shows the seasonal cycle of two provinces as example. Guangxi province is a typical rural province in China. About 60% of the province is covered by forest (www.forestry.gov.cn). Both DECSO v5 and v3b show high emissions from spring to summer. The $NO_x$ emissions estimated by DECSO v3b are only anthropogenic because biogenic emissions from

MEGAN are calculated internally by the CTM.  As we mentioned in section 3, DECSO v3b has a high background noise with an unrealistic seasonal cycle, which explain the difference with DECSO v5. We calculate that in the Guangxi province, the background noise of DECSO v3b contributes to a maximum of about 6 Gg per month in winter and 17 Gg per month in summer. By removing the background from DECSO v3b, the summer peak of emissions disappears. We see that monthly $NO_x$ emissions from March to September derived by DECSO v5 are much higher than the emissions of MIX in 2012. The

difference is probably due to biogenic emissions since MIX only includes anthropogenic emissions. DECSO v5 estimates



include both anthropogenic and biogenic emissions. In summer, the emissions derived by DECSO v5 are about 50% higher than in MIX. In Figure 5, we see that the biogenic emissions are 7-8% of anthropogenic emissions in summer based on the information given by MEGAN. However, the uncertainties of biogenic emissions derived by MEGAN are at least 300% (Guenther et al., 2006; 2012). Li et al. (2007) listed soil NO emission rates from different forests in the world including their

results of broadleaf and pine forests in Southern China. By using their range of emission rates, we calculate that the soil NO emissions in Guangxi in a range from 30 to 822 Gg N per year, the upper limit being 3.5 times higher than the yearly total $NO_x$ emissions derived by DECSO v5. This shows that the uncertainty in biogenic emissions from bottom-up emissions is quite high. DECSO v5 provide independent information about total $NO_x$ emissions in the region.

Anhui province is an example of an anthropogenic emission province. In this province $NO_x$ emissions derived by DECSO v5

are lower than MIX 2012. The peak that appears in March 2012 of DECOS v5 is probably due to high soil $NO_x$ emissions which are not included in MIX. Jaegle et al. (2005) concluded that globally soil $NO_x$ emissions can be as high as 20% of the total $NO_x$ emissions. Many studies demonstrate that both the usage of fertilizer and the increase of rainfall lead to significantly higher soil emissions (Bouwman et al., 2002; Cui et al., 2012; Ghude et al., 2010; Jaeglé et al., 2004). Liu et al. (2015) showed that lots of areas are over fertilized in Anhui. The climate annual report of Anhui

(http://www.ahqh.org.cn/product/yearqhmonitor.asp) shows that in March 2012 the precipitation is 50% higher than normal for the month and from May to July the province had a drought. This leads to the small peak of 10% higher $NO_x$ emissions in March 2012. DECSO v5 also shows a peak of $NO_x$ emissions in June 2013, which can be explained by the biomass burning in this month. The Globe Fire Emission Database (GFED) 4.0 (Giglio et al., 2013) shows large biomass burning areas in Anhui in May and June of 2013. Huang et al. (2012) pointed out that Anhui is one of the provinces having the

highest agricultural fire emissions with a peak in June. Stavrakou et al. (2016) conclude that the current estimates of agricultural fire emissions of post-harvest crop burning are much too low in China. These fires usually appear around the month of June, for which DECSO also shows higher emissions in Figure 6.

### 4.2 Ship track emissions near the Chinese coast

Figure 7 shows the yearly average emissions over the ocean in 2013 as derived with DECSO v5. We see a clear ship track

near the coast area from Guangzhou to Shanghai, which disperses in the Yellow Sea. Two high emission areas are at the edge of the Yangtze and Pearl River. In the Bohai Sea, at the coast of Shandong and Liaoning province, the emissions become stronger where ships are more concentrated on the route to and from Tianjin. A strong emission area appears at the coast of South Korea around the port of Pusan. Figure 8 shows the ship locations for one day based on Automatic Identification system (AIS) signals, the worldwide ship position monitor system (www.shipfinder.com). Most of the ships

are located close to the coast with a clear ship lane from Guangzhou to Shanghai, while ships become sparser in the Yellow sea. We see a good correlation of the high ship densities and the high emission areas derived by DECSO v5 shown in figure 7.

We select the Taiwan Strait area within the black rectangular shown in figure 7 to analyse the dependence of emissions on the distance to the coast. The total emissions for the selected area are about 49 Gg N year$^{-1}$ in DECSO v5 and 92 Gg N year$^{-1}$

in DECSO v3b. Figure 9 shows two peaks of the ship emissions in DECSO v5 as function of the distance from the coast. The maximum emissions are close to the coast within the territorial waters of 24 nautical miles (44 km). The ships along the coastline contribute the most to the emissions. The second peak of emissions is shown at a distance of about 90 km from the coast. As this distance we often find cargo ships on long distance routes from and to Shanghai or Tianjin. Note that the calculated background noise (emissions in the remote area as defined in section 3) are much lower than the ship emissions

shown here. The background noise of DECSO v5 indicated by the dashed blue line is as low as 0.017 Gg N per year per grid cell. The ship emissions derived by DECSO v3b show a similar pattern, but the values are two times higher due to the high background emissions (0.2 Gg N per year per grid cell).



When zooming in on Figure 8 (see the website http://www.shipfinder.com/), we distinguish two ship tracks in the selected area, which is also noticed by (Liu et al., 2016). One is within the territorial waters from 20 to 44 km from the coast, which we refer to as the territorial shipping. The other is within a distance of 44 to 160 km, which is referred to as the long distance shipping lane. Since it is the first time that the ship emissions near the Chinese coast are derived by satellite observations, we have no reference for the same area. However, because of the dispersion of international shipping routes into several branches, it is expected that emissions in our long distance shipping lane will be less than in the densely ship tracks between India and Singapore and in the Red Sea that have been studied by other authors. By dividing by the 800 km length of the ship lane, the emissions are about 0.025 Gg N km$^{-1}$ year$^{-1}$ in DECSO v5. Beirle et al. (2004) estimated the ship emissions from Sri Lanka to Indonesia were about 10 to 73 Gg N year$^{-1}$ (0.005 to 0.038 Gg N km$^{-1}$ year$^{-1}$) during 1996 to 2001. For the same area but from 2002 to 2007, Franke et al. (2009) estimated the ship emissions were about 90 Gg N year$^{-1}$ (0.047 Gg N km$^{-1}$ year$^{-1}$). Based on the study of Richter et al. (2004), the ship emissions for the Red Sea area are about 0.03 Gg N km$^{-1}$ year$^{-1}$ from 2002 to 2004. The magnitude of the shipping emissions over long distance shipping lane derived by DECSO v5 seems in agreement with these studies. The emissions derived from DECSO v5 in the Chinese territorial waters are much higher than for the long distance shipping, which seems a good reflection of the much higher density of ships in these waters.

## 5 Discussions and conclusions

We have improved the DECSO algorithm to calculate better emission estimates over remote emission regions. DECSO v5 estimates total NO$_x$ emissions instead of anthropogenic emissions to avoid the bias introduced by the uncertainties of biogenic emissions by MEGAN in the low emission areas. With a more solid approach of calculating error covariances of the sensitivity of NO$_2$ column observations to gridded NO$_x$ emissions, DECSO removes the artificial seasonality that is caused by daily calculation of error covariances. The background noise has been reduced in the new version with a factor of 10 to a level of less than 0.02 Gg N year$^{-1}$ per cell and a RMS (root mean square) error of 0.036 Gg N year$^{-1}$ per cell. The emission update constraint instead of an OmF filter enables DECSO to use 10% more observations and removes unrealistic emission fluctuations found in DECSO v3b. By changing the threshold value of the sensitivity matrix H, the errors caused by few observations on the edge of the emission plumes have been decreased.

Direct validation of emission estimates is not possible due to the lack of independent observations. However, we can roughly estimate the precision of the emission based on the year-to-year variability in the derived monthly emissions per grid cell in 2012 and 2013, since there is no significant trend in these two years (van der A et al., 2016). We also run DECSO v5 for these two years with different starting conditions with similar results. The precision is about 20% for each grid cell, while provincial total emissions have much better precision, which is less 2%.

With all improvements implemented in DECSO v5, it shows good agreement with the MIX inventory in the spatial distribution. Figure 10 shows an overview of NO$_x$ emission flux of each province from DECSO v5 and MIX in 2012. In some rural provinces (e.g. Guangxi, Yunan) NO$_x$ emissions of DECSO are about 30% higher than MIX. This is probably due to biogenic emissions that are not included in MIX. The NO$_x$ emissions in Guangxi are much higher than MIX in summer. NO$_x$ emissions of DECSO v5 are on average 30% lower than MIX and go up to 80% lower in the most of northern provinces of China. One possible reason for the low emissions in the North of China is the underestimation of NO$_2$ tropospheric columns of DOMINO v2 for specifically that area of the world. We find a significantly high number of negative retrievals over this area in the winter months, which indicates a negative bias in the retrievals. This is partly due to a bias in the high air mass factor for retrievals at large solar zenith angles, introduced by the multiple scattering in the radiance transfer model used in the calculation of the air mass factor in DOMINO (Lorente et al., 2016). Another reason for the underestimation of emissions can be the uncertainties of NO$_x$ sinks in the CTM used by DECSO for high latitudes. Stavrakou et al. (2013) showed that their NO$_x$ emission results are about 20% to 50% lower than MIX and they demonstrated the strong influence of



NO$_x$ loss uncertainties on top-down emission estimates. Mijling et al. (2013) discussed that the total anthropogenic NO$_x$ emissions in East China from different inventories have a large range, which can be as large as 40%. Except for the three most northern provinces the difference between DECSO v5 and MIX is within this range.

The relatively high R of 0.6 of seasonal emissions in anthropogenic emission provinces between DECSO v5 and MIX shows
that DECSO v5 is able to capture the seasonality of NO$_x$ emissions. However, the seasonality is not only dominated by anthropogenic emissions, biogenic emissions such as soil and fire emissions also have large impact on seasonality of emissions. Tie et al. (2006) showed that the total biogenic NO emissions in China are 50% of total anthropogenic emissions and mostly from agricultural soils and synchronized with large populations and human activity. Meanwhile agricultural fire emissions from crop burning after harvest activity are also quite high. NO$_x$ emissions of Anhui in 2012 and 2013 derived by
DECSO v5 show small peaks in different months and this may be caused by high soil and fire emissions in those months.

By reducing the background noise in DECSO, we see more clear ship tracks near the Chinese coastal areas, consistent with the ship locations. DECSO v5 reveals two shipping lanes in the Taiwan Strait: the territorial water shipping and the long distance shipping lane. The magnitude of the shipping emissions of long distance shipping lane derived by DECSO v5 is comparable with other studies on shipping emissions derived from satellite observations in other areas of the world with
similar ship density. The emissions in territorial water closer to the coast are much higher which is also shown by Fan et al. (2016). Shipping emissions from DECSO v5 can be further studied for seasonality and yearly trends and also for other parts of the seas near China.

In summary, DECSO v5 derives the total NO$_x$ emissions over East Asia with a very low background noise and reduces the artificial seasonal cycle over low emission regions compared to the previous versions. The seasonality of total emissions in
DECSO v5 is more realistic and it gives comprehensive information of emissions in East Asia. The emissions and their seasonality over remote areas like seas, deserts and grasslands are as easily derived as emissions over populated areas and can be updated on a regular basis. Combining the sector information of bottom-up inventories with the emissions calculated by DECSO, more accurate and timely inventories can be provided for the modelling community.

**Acknowledgments**

The research was part of the GlobEmission project funded and supported by the European Space Agency. We acknowledge Tsinghua University for providing the MIX emission inventory. We acknowledge IPSL/LMD, INERIS and IPSL/LISA in France for providing the CHIMERE model. We acknowledge the use of tropospheric NO$_2$ column data of DOMINO version 2 obtained from www.temis.nl

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



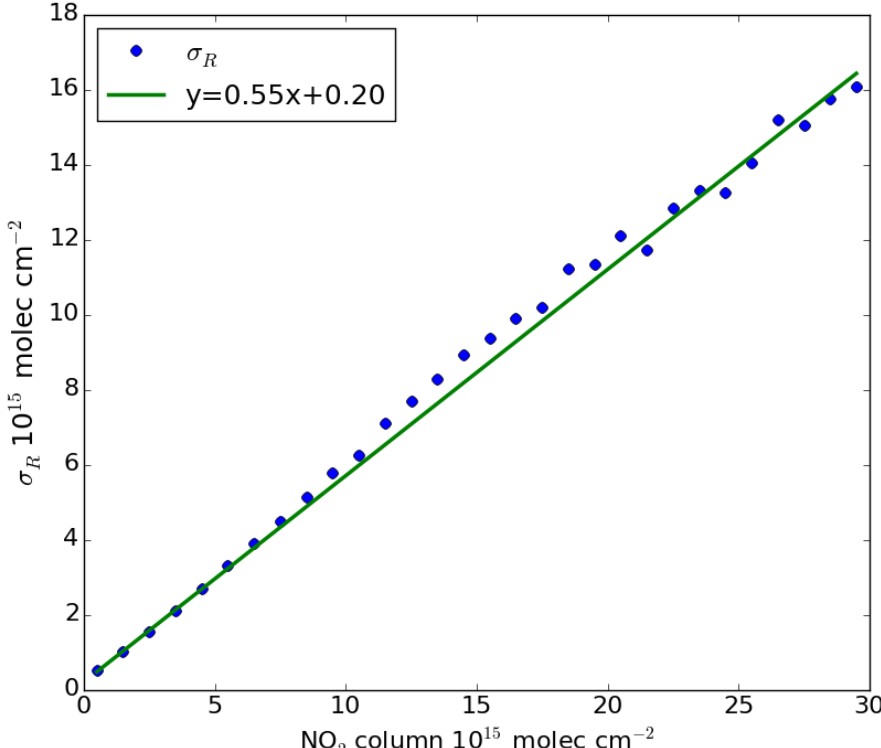

**Figure 1.** The relation between $\sigma_R$ and the average of the observed and modeled NO$_2$ column. The green line is the fit result.





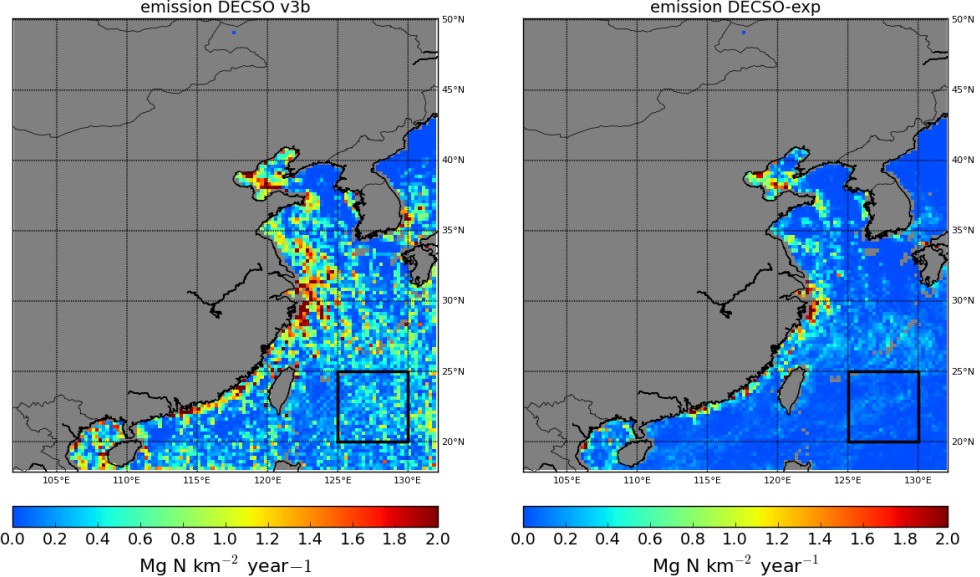

**Figure 2. The comparison of NO$_x$ emissions over the ocean derived with DECSO v3b (left) and DECSO-exp (right).**





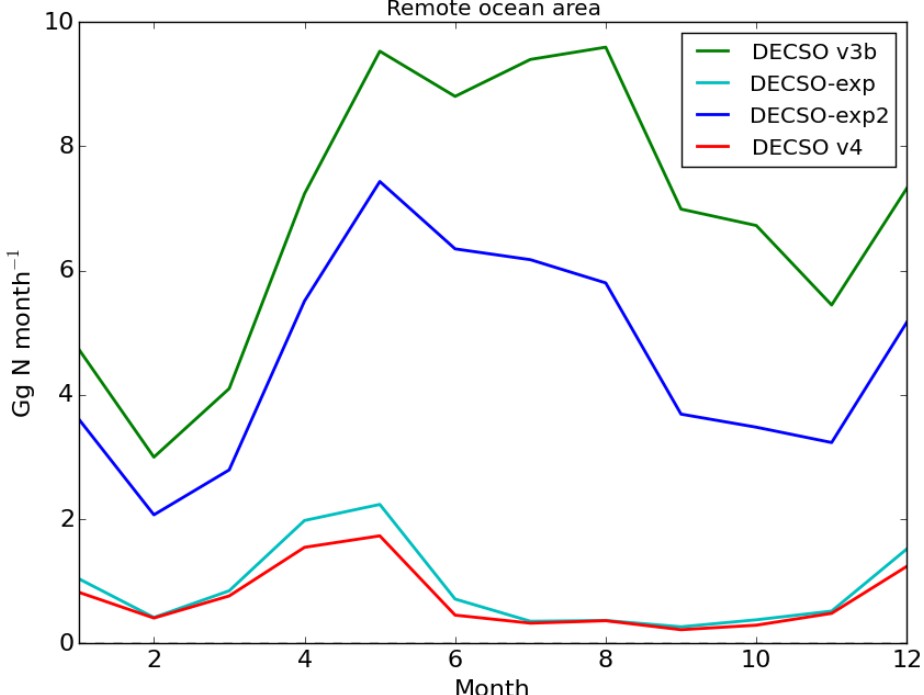

**Figure 3. The total monthly NO$_x$ emissions over a remote ocean area [20 to 25 °N, 125 to 130 °E] derived with DECSO v3b, DECSO-exp, DECSO-exp2 and DECSO v4.**





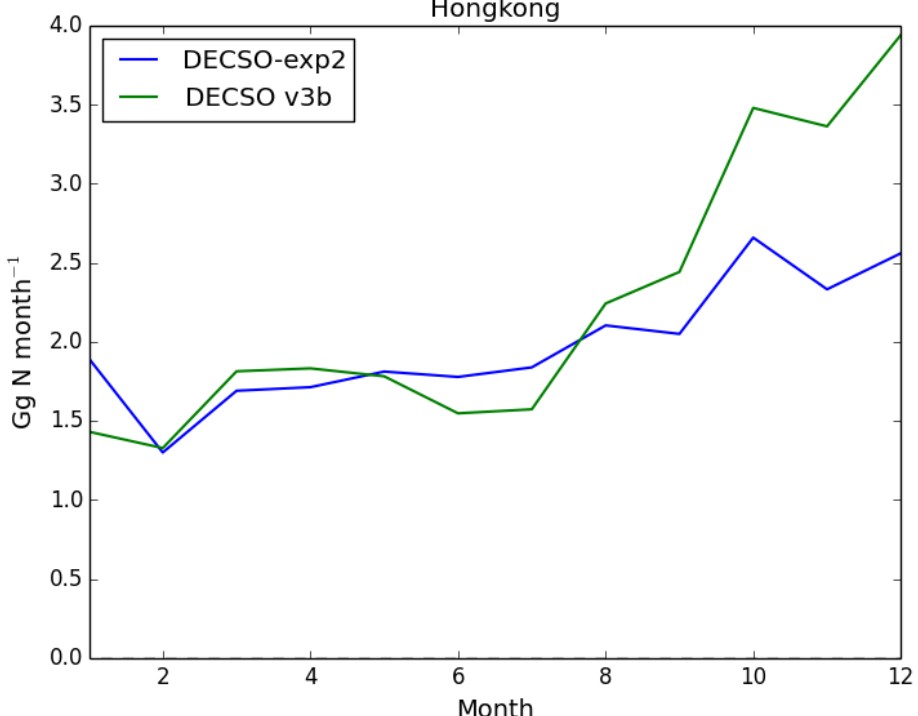

**Figure 4. The monthly emissions in Hongkong in 2013. DECSO-exp2 indicates the emissions derived using DECSO with the 3σ constraint. DECSO v3b indicates the emissions derived with the OmF filter.**



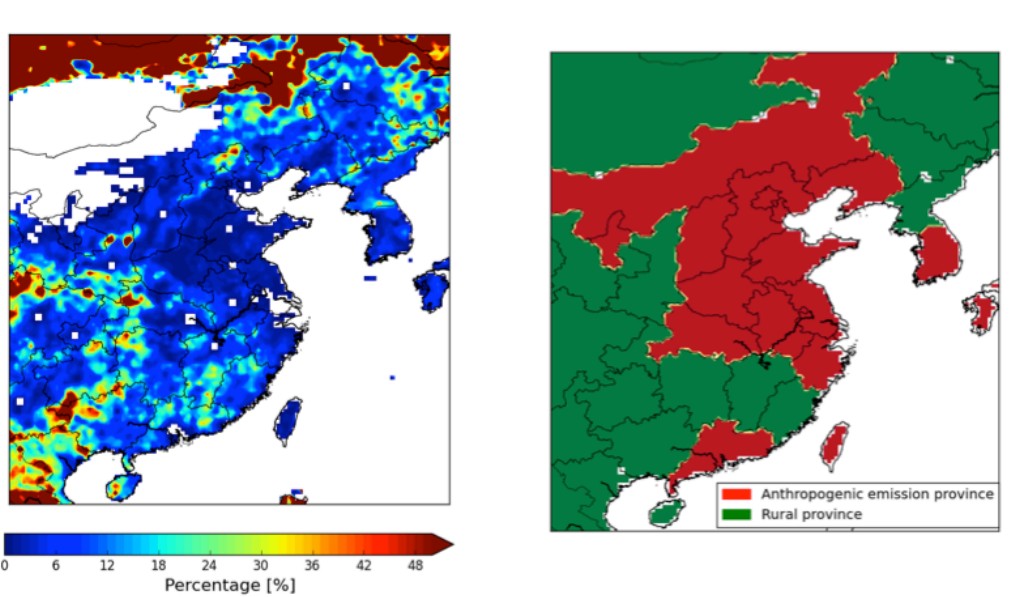

**Figure 5.** Ratio between biogenic and anthropogenic NO$_x$ emissions in summer (left) and the distribution of anthropogenic emission provinces and rural provinces (right). Biogenic emissions are from the MEGAN model used in CHIMERE. Anthropogenic emissions are from the MIX (2012) emission inventory.

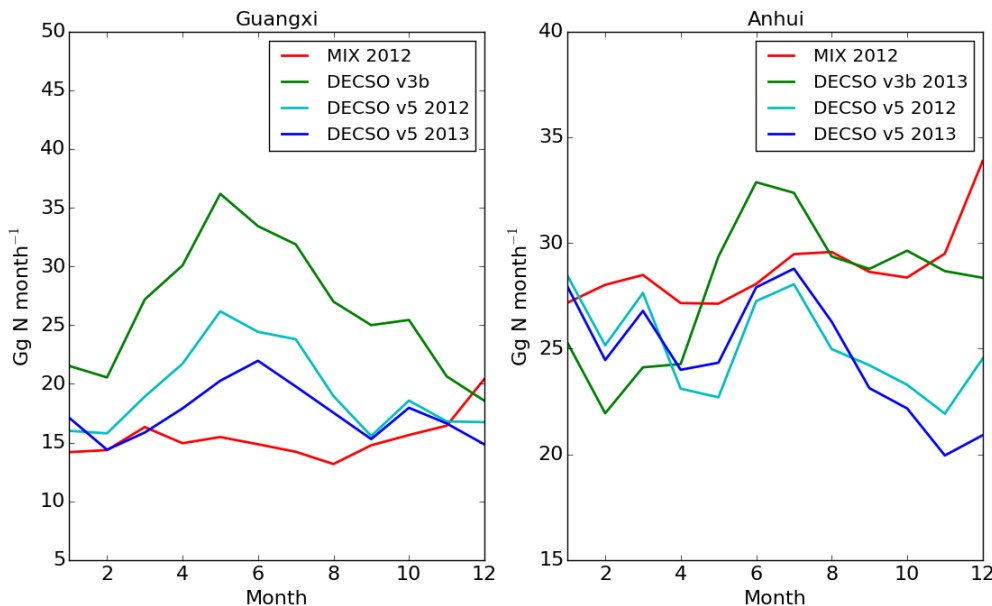

**Figure 6.** Monthly total emissions estimated by DECSO v5 (blue line) and DECSO v3b (green line) of Guangxi province (left) and Anhui province (right) in 2013. The cyan line and the red line represent monthly NO$_x$ emissions from DECSO v5 and the MIX inventory in 2012.





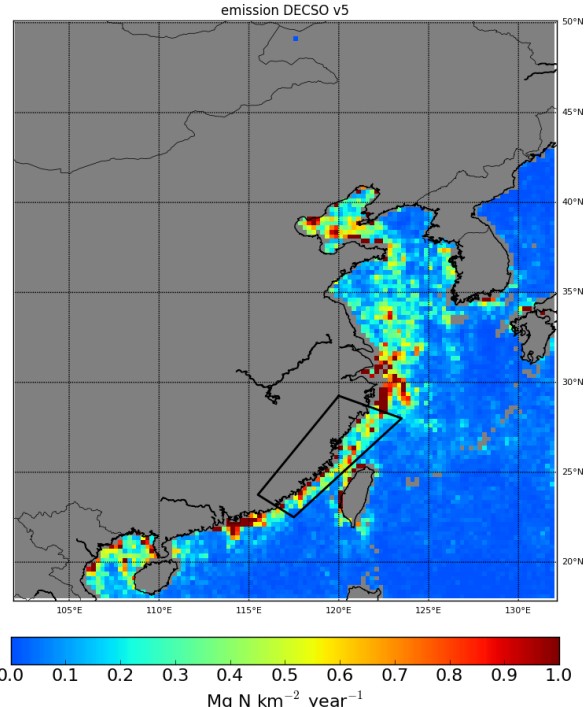

**Figure 7. Emission estimates derived with DECSO v5 over the ocean in 2013. The black quadrangle indicates the selected area for analyzing shipping emissions in the Taiwan Strait.**





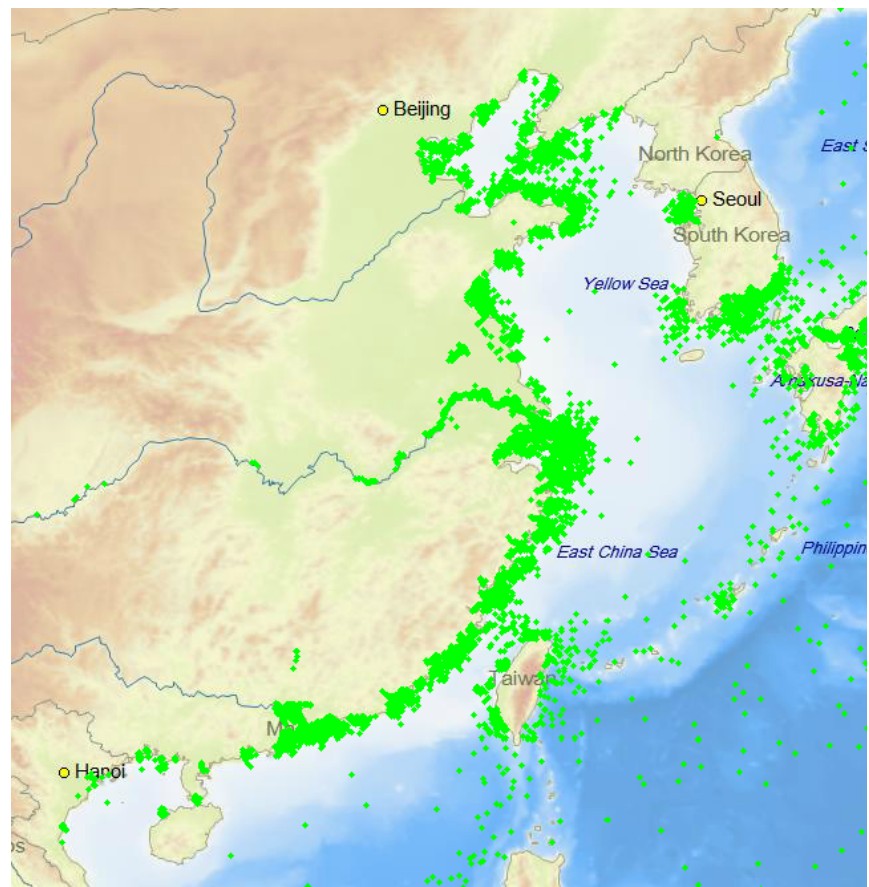

**Figure 8. Ship locations at 5 November 2015 from Automatic Identification System (AIS) information (taken from http://www.shipfinder.com/).**





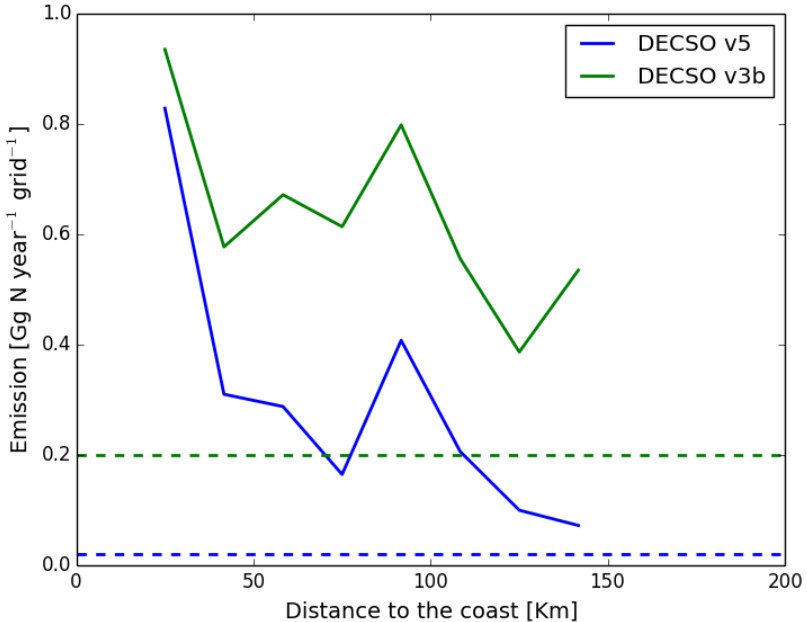

**Figure 9. The relation between NOₓ emissions over the ocean and the distance to the coastline in the Taiwan Street (the quadrilateral area in Figure 7). The blue dashed line is the background noise in DECSO v5 and the green dashed line in DECSO v3b.**

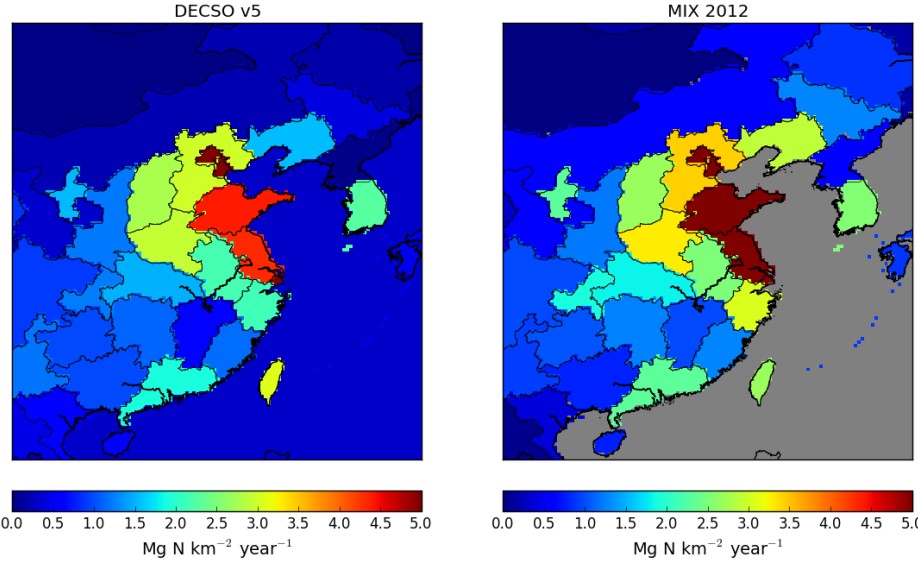

**Figure 10. Annual emissions on a provincial level derived with DECSO v5 and from the MIX (2012) inventory, used for China and outside China respectively. Emissions over sea are not included in the MIX inventory.**





**Table 1. Summary of main differences in various versions of DECSO**

| DECSO version | Description |
|---|---|
| v3b | The base version described in Ding et al 2015 |
| exp | Experimental version with improvement of the R matrix |
| exp2 | Experimental version with improvement of the 3σ constraint |
| v4 | Combined the improvements in exp and exp2 |
| v4log | Logarithmic version of v4 |
| v5 | Estimate total emissions instead of only anthropogenic emissions |

