# Peer review of "Space-based NOx emission estimates over remote regions improved in DECSO"

_Atmospheric Measurement Techniques, 2016_

## Referee Comment (RC1) · Anonymous Referee #1 · 10 Nov 2016

This paper by Ding et al. focuses on NOx emission estimates over remote regions using OMI observations and an improved version of the DESCO algorithm. The paper has many interesting aspects. I recommend publication after attention to the commenst below.

General comments:

Additional descriptions are needed in section 2 so that am independent author could reproduce the results from the information in the paper. For example, the prior emission inventory used and its error statistics (Pf) should be carefully enumerated. The authors refer to Mijling and van der A (2012) and to Ding et al. (2015) for detailed information, but the prior emissions used in those two studies are not identical.

The high resolution of the CHIMERE model system is a significant advantage over

other attempts to relate OMI NO2 measurements to emissions. However, the authors should acknowledge that there are known biases in the OMI product they are using that stem from low resolution elements of the retrieval. These will bias the resulting emissions by amounts that are comparable to the changes the authors derive. There will also be biases from the 25km resolution of the model that are not negligible especially for sources that are small compared to the grid scale. While it is not necessary to do new calculations, it is necessary that the paper discuss the results presetned in light of this related research.

Issues related to resolution effects on retrievals and models of NO2 are discussed in (among many other papers):

Heckel et al. Influence of low spatial resolution a priori data on tropospheric NO2 satellite retrievals, Atmos. Meas. Tech., 4, 1805-1820, doi: 10.5194/amt-4-1805-2011, 2011.

Kuhlmann et al. Development of a custom OMI NO2 data product for evaluating biases in a regional chemistry transport model, Atmos. Chem. Phys., 15, 5627-5644, doi:10.5194/acp-15-5627-2015, 2015.

Laughner, J. L., et al. Effects of daily meteorology on the interpretation of space-based remote sensing of NO2, Atmos. Chem. Phys. Discuss., doi:10.5194/acp-2016-536, in review, 2016.

McLinden et al. Improved satellite retrievals of NO2 and SO2 over the Canadian oil sands and comparisons with surface measurements, Atmos. Chem. Phys., 14, 3637-3656, doi:10.5194/acp-14-3637-2014, 2014.

Russell et al. A high spatial resolution retrieval of NO2 columns densities from OMI: method and evaluation, Atmos. Chem. Phys., 11, 8543-8554, doi:10.5194/acp-11-8543-2011, 2011.

Valin et al. Effects of model resolution on the interpretation of satellite NO2 observations, Atmos. Chem. Phys., 11, 11647-11655, doi:10.5194/acp-11-11647-2011, 2011.

Yamaji et al. Influence of model-grid resolution on NO2 vertical column densities over East Asia, J. Air. Waste. Manage., 64, 436-444, doi:10.1080/10962247.2013.827603, 2014.

Model error is an important aspect in emission inversion. As meteorological variables are not updated in DECSO, the sensitivity of observation to emissions (H) can suffer from model transport errors, which will bias the emission estimates. Discussion of the implication of model transport errors and its treatment on the emission inversion here.

The paper describes the resetting of the threshold of matrix H that represents the sensitivity of observation to emissions. The authors update the minimum value for H matrix elements from 0.05 to 0.1 hour. The motivation is that the H elements for observations located at the edge of the plume are usually small. Without explanations from a tracer transport perspective, I'm not convinced that setting a minimum is the appropriate approach to solve this problem. H elements calculated from the simplified 2D trajectories represent the contribution from model emission grid to the observations. Setting the 0.1 hour threshold could arbitrarily enhance this sensitivity for some emission points, and overcorrects the emissions which observations are not sensitive to. Some tests showing these effects are negligible and that a choice of 0.1 is optimal should be added.

The model assumption of persistent emissions is inconsistent with the behavior of biogenic, fire and lightning emissions. Additional discussion of this issue is needed.

For example, there is some knowledge of the mechanisms of biogenic emissions and models are available that represent processes. These processes vary strongly in repsonse to temerature and soil moisture. e.g. Hudman, et al.: A mechanistic model of global soil nitric oxide emissions: Implementation and space-based constraints, Atmos. Chem. Phys. Disc., 12, 3555-3594, 2012.

Do the derived biogenic emissions behave as expected in response to temperature or

rainfall?

It does not appear that lightning NOx emissions are represented in the model. Is it possible that the effects of lightning are interpreted as biogenic emissions?

Is it possible that fires are interpreted as biogenic emissions?

Details:

There are several other studies using Kalman filter and related methods to estimate NOx emissions The authors should cite them.

The equations, data and method should be provided for the calculation of soil emissions in Guangxi using emission rates from Li et al. (2007).

Figure 7: locations mentioned in section 4.2 should be marked in Figure 7 for readers who are not familiar with locations in Asia.

The authors should define "total emission" in this paper because it actually only includes anthropogenic and biogenic components. The general total emissions should include lightning NOx also. Suggestion is to rephrase it as total emissions from surface.

Support for the statement: "the errors caused by few observations on the edge of the emission plumes have been decreased" by updating H should be elaborated.

There are independent measurements from the national in situ observation network collected and maintained by the China National Environmental Monitoring Center (CNEMC).

―――――――――――――

---

## Referee Comment (RC2) · M. Wenig (Referee) · 30 Nov 2016

The manuscript "Space-based NOx emission estimates over remote regions improved in DECSO " by Ding et al. presents an improved version of the DESCO algorithm and its application to the Asian region and very interesting observations of ship tracks. The approach to derive emission inventories from satellite observations definitely address relevant scientific questions and is within the scope of AMT. The DESCO algorithm itself has been described in previous papers, but the improvement presented in this manuscript as well as the application to the Asian region is worth publishing as a separate paper. The study is generally suitable for publication in AMT, but needs some revisions as listed below.

Comments:

It is not clear to me how you can derive the uncertainty of 20% for the monthly grid cell emissions just by comparing 2 years. van der A et al. (2016) showed that the average NOx emission over Eastern China stays more or less constant in those two years, but also shows that the different provinces have their peak NOx emissions in different years, so the assumption of constant emissions might only hold on average but not for individual locations. What data are you comparing exactly, daily values or monthly? What starting conditions are you varying? Since with this approach you can only determine the precision but not the accuracy of your results, have you tried to determine systematic errors as well?

Would it be possible to use model data to test your algorithm? You could integrate the model output over height to simulate the satellite measurement, add some noise and then apply your retrieval technique. Of course you cannot determine how the model uncertainties affect your emission estimates, but at least you could compare improvement efforts to the algorithm.

You mention the precision of monthly emissions, but since you have the word 'daily' in your algorithm name, you might want to refer to the daily emission estimates.

In Sec. 2 you describe how you filter the observations, so it might be interesting to get to know how much data is left after that.

It might be helpful to better describe what the variables in Eq. 1 depend on. $e^{f(t)}$ e.g. looks like it only depends on the current day, but it depends mainly on the previous day, right? Unfortunately I couldn't find the time to read all the referenced papers with the more detailed algorithm description, so maybe more details are given there, but a more detailed in this paper might help. Does the observed NO2 column concentrations vector y only contains observations from the same day or also includes previous days?

The sentence starting in line 12 on page 5 is a little confusing. You might want to add 'forecasted' and 'measured' to the sentence "...the [] emissions of one day are equal to the [] emissions of the previous day" to be consistent with the equation. Please clarify

if the equation and the sentence are only true on average.

In addition to the correlation coefficients provided in lines 18ff on page 7, you could also mention slope and offset of a linear fit between modeled and measured NO2 columns, because a consistent over- or underestimation is equally important and doesn't show in the correlation coefficient.

Adding the locations of the cities mentioned in the text to Fig. 7 would make it easier for those not familiar with this area of the world to follow the description in Sec. 4.2.

Can you provide a correlation coefficient of the emission data from Fig. 7 with the ship location density from Fig. 8? It seems to be quite good for the quadrangle marked in Fig. 7 but not so much in the Yellow Sea where you detect some emissions as well, any idea why that is?

You write that changing the threshold value of the sensitivity matrix H reduced the errors and I'm wondering how you determined the optimal threshold.

---

## Author Comment (AC1) · 13 Jan 2017

We appreciate the referee #1 for giving valuable comments. We respond to each specific comment below and indicate where the changes we have made in the manuscript. The comments and questions from the referee are in blue italic font.

*General comments:*
*Additional descriptions are needed in section 2 so that an independent author could reproduce the results from the information in the paper. For example, the prior emission inventory used and its error statistics (Pf) should be carefully enumerated. The authors refer to Mijling and van der A (2012) and to Ding et al. (2015) for detailed information, but the prior emissions used in those two studies are not identical.*

The NOx emissions derived with DECSO become independent from the a priori emissions after the spin-up time of this algorithm, which is about 3 months. Different a priori emission inventories used in Mijling and van der A (2012) and Ding et al. (2015) have little influence on the final emission results.  To explain our method, we add the following sentences to line 6 of page 3:
"…..domain is taking into account. The transport is calculated using an ensemble of 150 isotopic 2-D trajectories for each grid cell. The inversion method used in DECSO is based on an extended Kalman filter. The extended Kalman filter is applied on emissions which uses a persistent emission forecast model. The emissions and their error covariance derived from DECSO are independent from the a prior emission inventory after a spin-up time of about 3 months."

On line 11 page 3, we add:
"..Range Weather Forecasts (ECWMF). The land use information we use for CHIMERE is from the GlobCover Land Cover database in 2009."

*The high resolution of the CHIMERE model system is a significant advantage over other attempts to relate OMI NO2 measurements to emissions. However, the authors should acknowledge that there are known biases in the OMI product they are using that stem from low resolution elements of the retrieval. These will bias the resulting emissions by amounts that are comparable to the changes the authors derive. There will also be biases from the 25km resolution of the model that are not negligible especially for sources that are small compared to the grid scale. While it is not necessary to do new calculations, it is necessary that the paper discuss the results presented in light of this related research.*

We agree with the referee that the $NO_2$ observations suffer from biases due to the limited resolution of the a priori information used in the retrieval cases with high NO2 concentration gradients (Heckel et al., 2011; Russell et al., 2011; Laughner et al., 2016). However, in China, there are few isolated emission sources with large concentration gradients. Therefore, several validation studies of the DOMINO v2 product with MAX-DOAS measurements show no significant biases in China on average (Irie et al., 2012; Wang et al., 2016). Ma et al. (2013) evaluated the gradient smoothing effect in the DOMINO v2 retrieval product in China and found no significant bias in winter and an upper limit for the bias in summer of 11-26%. Other studies show that the $NO_2$ observations suffer from biases in scenes with high aerosol concentrations in China (Lin et al., 2014; Kuhlmann et al., 2015; Chimot et al., 2016). Large biases coming from for example the high aerosol cases are avoided by the emission update constraint explained in section 3.2.

We add the discussion on line 25 page 9:

"Direct validation of emission estimates is not possible due to the lack of independent observations. The accuracy of the emission estimates depends largely on the accuracy of the NO2 satellite observations and of the CTM used in the inversion. Any random errors in the observations or the model are described in the Kalman Filter and result in an error estimate for the emissions. Biases are more difficult to quantify. A bias in the CTM can occur due to its limited resolution (Valin et al., 2011). In DECSO, the grid $NO_2$ columns simulated by the CTM are projected via a high resolution grid onto the footprint of satellite observations, which avoids the bias caused by different resolutions of the CTM and satellite observations. The $NO_2$ observations can suffer from biases in scenes with high aerosol concentrations (Lin et al., 2014; Kuhlmann et al., 2015; Chimot et al., 2016) or the limited resolution of the a priori information used in the retrieval cases with high $NO_2$ concentration gradients (Heckel et al., 2011; Russell et al., 2011; Laughner et al., 2016). In China, however, there are few isolated emission sources with large concentration gradients. Therefore, several validation studies of the DOMINO v2 product with MAX-DOAS measurements show no significant biases in China on average (Irie et al., 2012; Wang et al., 2016). Ma et al. (2013) evaluated gradient smoothing effect in the DOMINO v2 retrieval product in China and found no significant bias in winter and an upper limit for the bias in summer of 11-26%. Large biases coming from for example the high aerosol cases are avoided by the emission update constraint explained in section 3.2. "

*Model error is an important aspect in emission inversion. As meteorological variables are not updated in DECSO, the sensitivity of observation to emissions (H) can suffer from model transport errors, which will bias the emission estimates. Discussion of the implication of model transport errors and its treatment on the emission inversion here.*

We use the meteorological data from ECWMF with the time interval of 3h in the CTM CHIMERE. In the calculation of emission trajectory analysis, we interpolate the meteo data into half an hour time steps. The meteorological variables are updated. The errors of meteorological data only leads to random errors, which are covered by the Kalman filter. The threshold of H matrix described in section 3.4 helps to reduce the effect of the transport errors caused by errors in the meteorological data.
We explained this in the paper on page 3 line 6:
"….in which the transport of $NO_2$ over the model domain is taking into account. The transport is calculated using an ensemble of 150 isotopic 2-D trajectories for each grid cell. For the trajectory analysis, we use the operational meteorological forecast of the European Centre for Medium-Range Weather Forecasts (ECWMF) interpolated into half an hour time steps."

*The paper describes the resetting of the threshold of matrix H that represents the sensitivity of observation to emissions. The authors update the minimum value for H matrix elements from 0.05 to 0.1 hour. The motivation is that the H elements for observations located at the edge of the plume are usually small. Without explanations from a tracer transport perspective, I'm not convinced that setting a minimum is the appropriate approach to solve this problem. H elements calculated from the simplified 2D trajectories represent the contribution from model emission grid to the observations. Setting the 0.1 hour threshold could arbitrarily enhance this sensitivity for some emission points, and overcorrects the emissions which observations are not sensitive to. Some tests showing these effects are negligible and that a choice of 0.1 is optimal should be added.*

The H matrix plays an important role in the calculation of the Kalman Gain. The low value in the H matrix can lead to high values in the Kalman Gain analogue to the effect of

low Eigen values in any inversion. Therefore, the Kalman gain becomes sensitive to errors in the low values of H matrix elements. We have tested several threshold values of the sensitivity matrix H by comparing the results over some isolated hot spots. When the threshold is too high, we are running into numerical problems. 0.1 is the optimal choice based on these tests.

On line 33 page 6, we add:

"….. In this study we set the threshold value to 0.1 hour based on several tests using different threshold values"

*The model assumption of persistent emissions is inconsistent with the behavior of biogenic, fire and lightning emissions. Additional discussion of this issue is needed.*

The persistent emission model is for day to day evaluation. Emissions from any lasting changes are captured but with a delay which has been discussed in Ding et al. (2015). Fast, temporary changes of emissions (less than 1 day) cannot be captured or are underestimated by DECSO. Lightning emissions for example are not detected. Fires are usually detected since they last for several days.

*For example, there is some knowledge of the mechanisms of biogenic emissions and models are available that represent processes. These processes vary strongly in response to temerature and soil moisture. e.g. Hudman, et al.: A mechanistic model of global soil nitric oxide emissions: Implementation and space-based constraints, Atmos. Chem. Phys. Disc., 12, 3555-3594, 2012.*
*Do the derived biogenic emissions behave as expected in response to temperature or rainfall?*

We could refine our persistency model with such information, but this would mean that we add a priori information based on land use, temperature and soil moisture. This would add additional complications and we still would miss for example the commissioning of a new power plant. Therefore, we prefer using a very simple model without a priori information that is able to follow changes on a time scale of days or longer.

In Mijling et al. (2013), we see that the periodicity of total NOx emissions in Mongolia follow the rain/temperature cycle.

*It does not appear that lightning NOx emissions are represented in the model. Is it possible that the effects of lightning are interpreted as biogenic emissions?*
*Is it possible that fires are interpreted as biogenic emissions?*

We are not able to distinguish the emissions of different source categories; so all emissions derived are represented as surface total emissions. If the emissions changes are due to lightning, in DECSO, it is considered to the surface level.

We emphasize it by changing the text on line 9 page 8:

" DECSO v5 provides independent information about total surface $NO_x$ emissions in the region since it is difficult to distinguish source types of the emissions.

*Details:*
*There are several other studies using Kalman filter and related methods to estimate NOx emissions The authors should cite them.*

We add the reference in the introduction on page 1 line 33:

"…This algorithm takes the transport of NO$_x$ from its source into account by including 2-D trajectory analysis in the sensitivity of NO$_2$ concentration on NO$_x$ emissions. Other NOx emission studies based on a Kalman Filter often use the decoupled direct method to calculate the sensitivity (Napelenok et al., 2008; Tang et al., 2013) or an ensemble Kalman Filter (Miyazaki et al., 2012)."

*The equations, data and method should be provided for the calculation of soil emissions in Guangxi using emission rates from Li et al. (2007).*

We calculate the emissions by multiplying emission rates with the area of the forests in the province.

We change the text on page 7 line 34:

"About 60% of the province is covered by forest ([www.forestry.gov.cn](www.forestry.gov.cn)), which is about 14×10$^5$ km$^2$."

On page 8 line 8, we change the text into:

"By using their range of emission rates, we calculate that the soil NO emissions in Guangxi in a range from 30 to 822 Gg N per year by multiplication with the forest area, the upper limit being…"

*Figure 7: locations mentioned in section 4.2 should be marked in Figure 7 for readers who are not familiar with locations in Asia.*

We add markers in Figure 7 mentioned in section 4.2 and change the figure 7 and change Figure 7 in the paper.

[Figure]

*The authors should define "total emission" in this paper because it actually only includes anthropogenic and biogenic components. The general total emissions should include lightning NOx also. Suggestion is to rephrase it as total emissions from surface.*

We define the total emissions by changing the text on line 4 page 7:

"By excluding biogenic NO$_x$ emissions in CHIMERE, we derive the total surface NO$_x$ emissions from satellite observations. The total emissions from surface include anthropogenic, biogenic and fire emissions. "

*Support for the statement: "the errors caused by few observations on the edge of the emission plumes have been decreased" by updating H should be elaborated.*

See the discussion above

*There are independent measurements from the national in situ observation network collected and maintained by the China National Environmental Monitoring Center (CNEMC).*

There are indeed independent measurements from the national in-situ observation network. Since the validation of emissions with direct observations is difficult unless a CTM is involved, the results are highly related to the choice of the CTM. We will address the validation of emissions in future work.

References

Chimot, J., Vlemmix, T., Veefkind, J. P., de Haan, J. F., and Levelt, P. F.: Impact of aerosols on the OMI tropospheric $NO_2$ retrievals over industrialized regions: how accurate is the aerosol correction of cloud-free scenes via a simple cloud model?, Atmos. Meas. Tech., 9, 359-382, 2016.

Ding, J., van der A, R. J., Mijling, B., Levelt, P. F., and Hao, N.: $NO_x$ emission estimates during the 2014 Youth Olympic Games in Nanjing, Atmos. Chem. Phys., 15, 9399-9412, 2015.

Heckel, A., Kim, S. W., Frost, G. J., Richter, A., Trainer, M., and Burrows, J. P.: Influence of low spatial resolution a priori data on tropospheric $NO_2$ satellite retrievals, Atmos. Meas. Tech., 4, 1805-1820, 2011.

Irie, H., Boersma, K. F., Kanaya, Y., Takashima, H., Pan, X., and Wang, Z. F.: Quantitative bias estimates for tropospheric $NO_2$ columns retrieved from SCIAMACHY, OMI, and GOME-2 using a common standard for East Asia, Atmos. Meas. Tech., 5, 2403-2411, 2012.

Kuhlmann, G., Lam, Y. F., Cheung, H. M., Hartl, A., Fung, J. C. H., Chan, P. W., and Wenig, M. O.: Development of a custom OMI $NO_2$ data product for evaluating biases in a regional chemistry transport model, Atmos. Chem. Phys., 15, 5627-5644, 2015.

Laughner, J. L., Zare, A., and Cohen, R. C.: Effects of daily meteorology on the interpretation of space-based remote sensing of $NO_2$, Atmos. Chem. Phys., 16, 15247-15264, 2016.

Lin, J. T., Martin, R. V., Boersma, K. F., Sneep, M., Stammes, P., Spurr, R., Wang, P., Van Roozendael, M., Clémer, K., and Irie, H.: Retrieving tropospheric nitrogen dioxide from the Ozone Monitoring Instrument: effects of aerosols, surface reflectance anisotropy, and vertical profile of nitrogen dioxide, Atmos. Chem. Phys., 14, 1441-1461, 2014.

Ma, J. Z., Beirle, S., Jin, J. L., Shaiganfar, R., Yan, P., and Wagner, T.: Tropospheric $NO_2$ vertical column densities over Beijing: results of the first three years of ground-based MAX-DOAS measurements (2008–2011) and satellite validation, Atmos. Chem. Phys., 13, 1547-1567, 2013.

Mijling, B., van der A, R. J., and Zhang, Q.: Regional nitrogen oxides emission trends in East Asia observed from space, Atmos. Chem. Phys., 13, 12003-12012, 2013.

Miyazaki, K., Eskes, H. J., and Sudo, K.: Global $NO_x$ emission estimates derived from an assimilation of OMI tropospheric $NO_2$ columns, Atmospheric Chemistry and Physics, 12, 2263-2288, 2012.

Napelenok, S. L., Pinder, R. W., Gilliland, A. B., and Martin, R. V.: A method for evaluating spatially-resolved $NO_x$ emissions using Kalman filter inversion, direct sensitivities, and space-based $NO_2$ observations, Atmos. Chem. Phys., 8, 5603-5614, 2008.

Russell, A. R., Perring, A. E., Valin, L. C., Bucsela, E. J., Browne, E. C., Wooldridge, P. J., and Cohen, R. C.: A high spatial resolution retrieval of $NO_2$ column densities from OMI: method and evaluation, Atmos. Chem. Phys., 11, 8543-8554, 2011.

Tang, W., Cohan, D. S., Lamsal, L. N., Xiao, X., and Zhou, W.: Inverse modeling of Texas $NO_x$ emissions using space-based and ground-based $NO_2$ observations, Atmos. Chem. Phys., 13, 11005-11018, 2013.

Valin, L. C., Russell, A. R., Hudman, R. C., and Cohen, R. C.: Effects of model resolution on the interpretation of satellite $NO_2$ observations, Atmos. Chem. Phys., 11, 11647-11655, 2011.

Wang, Y., Beirle, S., lampel, J., Koukouli, M., De Smedt, I., Theys, N., Li, A., Wu, D., Xie, P., Liu, C., Van Roozendael, M., and Wagner, T.: Validation of OMI, GOME-2A and GOME-2B tropospheric $NO_2$, $SO_2$ and HCHO products using MAX-DOAS observations from 2011 to 2014 in Wuxi, China, Atmos. Chem. Phys. Discuss., 2016, 1-50, 2016.

---

## Author Comment (AC2) · 13 Jan 2017

We thank Mark Wenig for his positive review and valuable comments. We respond to each specific comment below. The comments and questions from the referee are in italic font and blue color.

*It is not clear to me how you can derive the uncertainty of 20% for the monthly grid cell emissions just by comparing 2 years. Van der A et al. (2016) showed that the average NOx emission over Eastern China stays more or less constant in those two years, but also shows that the different provinces have their peak NOx emissions in different years, so the assumption of constant emissions might only hold on average but not for individual locations. What data are you comparing exactly, daily values or monthly? What starting conditions are you varying? Since with this approach you can only determine the precision but not the accuracy of your results, have you tried to determine systematic errors as well?*

The uncertainty we present here is based on the statistics. We first calculated the difference in monthly emissions of each grid cell by comparing the emissions of the two years. The domain includes 15609 grid cells, which means we have 15609 x 12 samples. The average difference is about 20%, which includes the uncertainty and the trend (which is small). We compared monthly emissions of each grid cell in 2012 by running DECSO v5 with different initial emission inventories and starting years. The difference is less than 20% as well. In the same way, we calculated the difference of each province from two different runs (30 x 12 samples), which is less than 2%. We only calculated the precision in this study. But we plan to analyze the accuracy in more detail by comparing different emission inventories in a follow-up study.

We change the text on line 27 page 9:

"... However, we can roughly estimate the precision of the emission based on the year-to-year variability in the derived monthly emissions per grid cell in 2012 and 2013, since there is no significant trend in these two years (van der A et al., 2016). We calculate the average difference in monthly emissions between 2012 and 2013 over all grid cells (15609 grid cells), which is about 20%. We verify the result by comparing the derived monthly emissions for 2012 from DECSO with a run with a different initial emission inventory and starting year. We conclude that the precision is about 20% for each grid cell. We do the same calculations on a provincial level and find that the provincial monthly emissions have a much better precision of less than 2%. "

*Would it be possible to use model data to test your algorithm? You could integrate the model output over height to simulate the satellite measurement, add some noise and then apply your retrieval technique. Of course you cannot determine how the model uncertainties affect your emission estimates, but at least you could compare improvement efforts to the algorithm.*

We agree that this is a very good strategy to monitor the improvements. This has been done in section 5 of Mijling and van der A (2012). However, we use a different approach for the new version. We tested our improvement by using one grid cell box model with artificial observations. We haven't mentioned this in the paper. Earlier results of this box model were also mentioned in Mijling and van der A (2012).

*You mention the precision of monthly emissions, but since you have the word 'daily' in your algorithm name, you might want to refer to the daily emission estimates.*

The DECSO algorithm derives daily emissions because we use satellite observations on a daily basis. The response in emission updates, however, depends on the number of daily

observations (after filtering), and the retrieval error. Current retrieval products from instruments like OMI and GOME-2 lack the spatial/temporal resolution nor the accuracy to capture strong day-to-day changes in emission. Monthly emissions have an almost full coverage of the whole domain and are our final product. That's why we analyze the precision of monthly emissions.

To emphasize this, we add the following sentence on line 9 page 7:

"...over East Asia. The final results are monthly emissions for this period. As we showed ..."

*In Sec. 2 you describe how you filter the observations, so it might be interesting to get to know how much data is left after that.*

After the filtering, we have about 2000 observations per day over the domain of East Asia. Note that this has a strong seasonal cycle due to cloud climatology and snow (which for example lowers the response time in winter and rainy seasons.)

We add this information on line 20 page 3:

"... the retrieval product. After this filtering, we typically have about 2000 observations per day for each domain."

*It might be helpful to better describe what the variables in Eq. 1 depend on. eˆf(t) e.g. looks like it only depends on the current day, but it depends mainly on the previous day, right? Unfortunately I couldn't find the time to read all the referenced papers with the more detailed algorithm description, so maybe more details are given there, but a more detailed in this paper might help. Does the observed NO2 column concentrations vector y only contains observations from the same day or also includes previous days?*

Yes, $e^f(t)$ depends on the previous day. We assume a persistent emission model, which means the forecasted emission of the current day is equal to the analysis of the emission from the previous day. The NO2 column concentration vector only contains the observations from the current day. To make it more clear, we change the text below at line 30 page 3:

"... the observed $NO_2$ column concentrations **y** at time t and the forecasted ..."

We also change the text at line 34 page 3:

"....satellite footprint. $\mathbf{e}^f(t)$ is, equal to the analysis of the emissions from the previous day, following a persistent emission model. The Kalman..."

We add more description about DECSO in Section 2 (page 3 line 5):

"The essential part of DECSO is the calculation of the sensitivity of the $NO_2$ column concentrations (on a footprint of the satellite) to the gridded $NO_x$ emissions, in which the transport of $NO_2$ over the model domain is taking into account. The transport is calculated using an ensemble of 150 isotopic 2-D trajectories for each grid cell. For the trajectory, we use the operational meteorological forecast of the European Centre for Medium-Range Weather Forecasts (ECWMF) interpolated into half an hour time steps. The inversion method used in DECSO is based on an extended Kalman filter. The emissions and their error covariance derived from DECSO are independent from the a prior emission inventory after a spin-up time of about 3 months."

*The sentence starting in line 12 on page 5 is a little confusing. You might want to add 'forecasted' and 'measured' to the sentence ". . .the [] emissions of one day are equal to the [] emissions of the previous day" to be consistent with the equation. Please clarify if the equation and the sentence are only true on average.*

The equation is based on an assumption that emissions are the same from day to day within a certain error margin. We could refine our persistency model with more information on biogenic emissions, but this would mean that we add a priori information based on land use, temperature and soil moisture. This would add additional complications and we still would miss for example the building of a new power plant. Therefore, we prefer using a very simple model without a priori information that is able to follow changes on a time scale of days or longer.

We change the sentence into:
"In the DECSO algorithm, we use a persistent emission model, which assumes that the forecasted emissions of the current day are equal to the analysis of the emission of the previous day."

*In addition to the correlation coefficients provided in lines 18ff on page 7, you could also mention slope and offset of a linear fit between modeled and measured NO2 columns, because a consistent over- or underestimation is equally important and doesn't show in the correlation coefficient.*

We calculate the correlation coefficients with satellite observations on the spatial distribution to show locations of emissions of DECSO v5 are more precise than of the old version. The slopes with and without fixed offset of a linear fit between modeled and measured NO2 columns are both improved in DECSO v5. However, we decide not to present the slope and offset in the paper because the datasets of two versions are not comparable, since we calculate the coefficient following the two steps below:
1. We used the daily modeled columns which are projected in to the footprint of satellite observations and applied with Average Kernel. This dataset is from the assimilation process.
2. We regridded both daily modeled and observed NO2 columns used in the assimilation process on the footprint into the grid cells and calculate the monthly, seasonal and yearly average to calculate the correlation coefficient of spatial distribution. Note that the observation data used here are the data used in the DECSO algorithm, which means the observation data are not exactly the same in two versions. DECSO v3b uses an OmF satellite data filter and DECSO v5 doesn't have the filter but an emission update constraint. So all the outliers of modeled and satellite columns are filtered out if the OmF are large in DECSO v3b.

The further validation of emissions with both in-situ and satellite observations is future work.

*Adding the locations of the cities mentioned in the text to Fig. 7 would make it easier for those not familiar with this area of the world to follow the description in Sec. 4.2.*

We add the locations of the cities and provinces in Fig 7.
We change Figure 7 in the paper.

[Figure]

NOx emissions (DECSO v5)

Provinces:
   1 Liaoning
   2 Shandong
   3 Guangzhou
Cities:
   4 Busan
   5 Tianjin
   6 Shanghai
Rivers and Seas:
   7 Bohai Sea
   8 Yangtze River
   9 Pearl River
   10 Yellow sea

0.0  0.1  0.2  0.3  0.4  0.5  0.6  0.7  0.8  0.9  1.0
$Mg\ N\ km^{-2}\ year^{-1}$

*Can you provide a correlation coefficient of the emission data from Fig. 7 with the ship location density from Fig. 8? It seems to be quite good for the quadrangle marked in Fig. 7 but not so much in the Yellow Sea where you detect some emissions as well, any idea why that is?*

Unfortunately, we don't have the underlying data of figure 8. Therefore we cannot calculate the correlation coefficient of the emission data from Fig 7 with the ship location. We filter the grid cells, which include any part of the land. The ships showing in figure 8 in the yellow sea are mainly at the coast and inland water. We add the description of the filtering on line 24 page 8:
"… derived with DECSO v5. We filter out the grid cells including any part of the land because we cannot distinguish shipping emissions from land-based emissions. …"

We add the following sentence on line 32 page 8:
"…by DECSO v5 shown in figure 7. Many of the ship locations in figure 8 are close to the coast or on inland water and therefore are not visible in figure 7."

*You write that changing the threshold value of the sensitivity matrix H reduced the errors and I'm wondering how you determined the optimal threshold.*

We have tested several threshold values of the sensitivity matrix H by comparing the results over some isolated hot spots. When the threshold is too high, we are running into numerical problems. 0.1 is the optimal choice based on these tests.
On line 33 page 6, we add:
"….. In this study we set the threshold value to 0.1 hour based on several tests using different threshold values"

References

Mijling, B., and van der A, R. J.: Using daily satellite observations to estimate emissions of short-lived air pollutants on a mesoscopic scale, Journal of Geophysical Research: Atmospheres, 117, 10.1029/2012JD017817, 2012.

van der A, R. J., Mijling, B., Ding, J., Koukouli, M. E., Liu, F., Li, Q., Mao, H., and Theys, N.: Cleaning up the air: Effectiveness of air quality policy for $SO_2$ and $NO_x$ emissions in China, Atmos. Chem. Phys. Discuss., 2016, 1-18, 10.5194/acp-2016-445, 2016.